# Micro Aspheric Convex Lenses Fabricated by Precise Scraping

**DOI:** 10.3390/mi13050778

**Published:** 2022-05-15

**Authors:** Meng-Ju Lin

**Affiliations:** Department of Mechanical and Computer-Aided Engineering, Feng Chia University, Taichung 407102, Taiwan; mengjlin@fcu.edu.tw; Tel.: +886-4-24517250-3554

**Keywords:** microsurface scraping, cutter-path planning, aspherical lenses, microshaper

## Abstract

An easy, fast, inexpensive, and simple method utilizing a microshaper with a very small knife nose is used to fabricate microconvex aspherical lenses. The microshaper is mounted on a computer numerical control (CNC) machine. To achieve an accurately designed profile of the lens surface, a cutter-path planning algorithm with compensation for knife interference is developed. Exerting this algorithm in CNC machining, the microconvex aspheric surface is precisely scraped. To verify the precise machining of the cutter path planning algorithm, three aspheric surfaces of conic sections (ellipsoid, paraboloid, and hyperboloid) are successfully fabricated. The profiles scraped by the microshaper agree well and precisely with the designed theoretical conic section curve. Using a simple polishing method to make the machined surface smoother, the roughness is reduced from 143 and 346 nm to 52 and 44 nm for the path line direction and its transverse direction, respectively. The micro-aspherical lenses have moderate machining properties using a simple polishing method. The results show that the designed profiles of micro-aspheric convex lenses can be machined precisely and efficiently by the microshaper with the cutter-path planning algorithm developed in this work. From the image comparison formed by the aspherical and spherical microlenses, the aspherical lenses provide a better image. It is feasible that the designed profile of the micro-aspherical lenses with specific functions could be machined using the cutter-path planning algorithm developed in this work.

## 1. Introduction

For the fast-growing and developing applications in sensing, communication, detection, etc., micro-optoelectromechanical systems (MOEMSs) [1,2] have become an important category of micro-electromechanical systems (MEMS). For their physical characteristics of tracking, collimating, and coupling lights [3,4,5], microlenses have become essential and important components used in MOEMSs. They are utilized in imaging devices such as cameras [6], confocal microscopes [7], and displays [8,9] due to their light focusing and collimation ability. Microlenses have also developed more specific and novel applications. The microlenses are important components used in Mirau interferometers [10], scanners [11,12], integrated photonic platform [13], vertical-cavity surface-emitting laser beam shaping [14], telecommunications [15], data storage [16], biodetection [17], etc. Due to their important role in MOEMSs, microlenses have recently been more widely discussed.

To fabricate microlenses, silicon-based micromachining is often used for its compatibility with integrated circuits (IC) and other components of MEMS. Based on silicon-based micromachining, several techniques are developed. To fabricate microlenses, their surfaces can be made by lithography with LIGA-like processes using excimer laser micromachining [18], combining reactive-ion etching (RIE) with modified parameters [19] and the reflow method, solidification of injecting droplets [20], heating photoresist to liquid phase and becoming curved by its capillary force [21], and using a cross-linked network of photoactive polymers [22]. Among such fabricating methods, reflow techniques have become an important process in the fabrication of microlenses. Therefore, reflow is widely used and well-developed in microlenses’ fabrication using silicon-based micromachining. Through the reflow of different materials (photoresist [23], silicon dioxide [24], etc.) combined with other silicon-based micromachining processes, microleneses and micro-optical and communicating devices can be fabricated. For example, microlenses can be fabricated by combining the dose-modulated lithography and reflow process [25]. Microleneses used in the antenna-integrated heterodyne array with high frequency (THz) can be made by reflow with RIE [26,27]. The microlenses can also be fabricated by reflow with nanoimprint [28,29] or reflow combined with ultraviolet nanoimprint lithography and replica mold processes [30], etc. For the afore-mentioned silicon-based micromachining methods, they focus on the applications of microlenses. Although the lenses with quasi-spherical shapes can be fabricated according to physical characteristics, by using these methods it is hard to control the precise profile of the lenses. In addition, non-silicon-based micromachining can be used to fabricate microlenses. Using a microshaper mounted on a three-axis computer numerical control (CNC) machine, the spherical profiles of the microlenses can be machined [31]. The results show that the microlenses have precise profiles and focal lengths. Traditionally, convex lenses often have spherical surfaces for their easily fabricating and high yield. Therefore, the microlenses are also often made of spherical or quasi-spherical profiles. However, spherical lenses have aberration problems and could cause blurry images. Instead of spherical lenses, the aspherical lenses can compensate for aberration for better images. To fabricate micro-aspherical lenses using silicon-based micromachining, the methods include electro-wetting [32,33,34,35,36,37], stamping [37,38], molding and hot forming [39,40], hydraulic control [41], and electrostatic-force-modulation [42,43,44]. It is known that silicon-based micromachining has its advantages in batch fabrication, mass production, and integration with electronic components. The afore-mentioned works presented aspherical lenses that can be fabricated using silicon-based micromachining. However, their aspherical profiles are formed by using fabrication parameters and applying voltages. They have no designed functions for the lens profiles. Therefore, the reported methods to fabricate aspherical microlenses cannot control the precise shape and profile of the lenses. It is hard to precisely fabricate design profiles with a specific function of the micro-aspherical lens using silicon-based micromachining. However, microshapers mounted on CNC machines with cutter-path planning can scrape precise spherical profiles [31]. Furthermore, this simple, easy, inexpensive, and quick non-silicon-based micromachining method can make prototypes that need to be cyclically modified without preparing photomasks for lithography and some processes need a vacuum environment. This method can save fabricating time and costs for cyclic modifications and can help prototypes be tested and designed. In this study, the microshaper mounted on a CNC machine is used to fabricate micro-aspherical lenses. For machining the precisely designed profiles of micro-aspherical lenses, the algorithm of cutter-path planning is significant. In this work, the cutter-path planning algorithm is developed. The compensation algorithm to avoid knife interference is also derived. With the cutter-path planning algorithm, the aspherical microlenses of conic surfaces are precisely and successfully fabricated to verify the feasibility of the algorithm. The machined profiles by microscraping agree well with the design profile equations. Moreover, the roughness, which is important for optical properties, can be diminished using simple polishing [31]. In this work, the micro-aspherical lenses have moderate roughness of 52 and 44 nm for path line direction and path line transverse direction, respectively. Furthermore, the micro aspherical lenses have a better image. Therefore, precisely machining a profile with a specific function using the algorithm of cutter-path planning developed in this work is feasible.

## 2. Materials and Methods

In recent decades, machining using CNC machines has been well-developed. The small or tiny components with micrometer dimensions can be fabricated using CNC precision machining [1]. It is known that the most significant advantage of CNC machining is its ability to manufacture complex curved surfaces. To manufacture three-dimensional curved surfaces on a CNC machine, ball-end milling is an often-used method. It can efficiently machine complex profiles. The complex, curved three-dimensional structures can be executed by the synchronous multi-axis movement of the ball-end milling of the CNC machines [45,46,47]. However, some synchronous multi-axis CNC machines are expensive. For easy and inexpensive machining, the three-axis CNC machine is chosen to fabricate microlenses. Precisely machining complex curved surfaces could be achieved by cutter-path planning [48,49,50,51,52,53,54,55,56,57]. For cutter-path planning methods, free-form surfaces machined by CNC milling are well-investigated and have good performance [58]. It is chosen in this work as the cutter-path-planning method. However, due to the limit of the knife radius in ball-end milling, fabricating profiles of microcurved surfaces is a challenge for traditional CNC milling. Therefore, in order to make microcurved surfaces on a CNC machine, finding a specific knife is important. In this work, a microshaper is used to fabricate a microsurface. The microshaper has a small knife nose. Due to the small nose size, the microshapers can remove tiny material volumes and manufacture microcurved surfaces. This microshaper has a small nose radius of 85 μm as shown in Figure 1. The material of this microshaper is tungsten steel (tungsten carbide), which has adequate hardness, stiffness, and wear resistance [59,60]. Therefore, ductile materials such as metal, polymer, plastic, etc., can be scraped by this microshaper. Furthermore, if coating a diamond layer on the nose of the knife tip, this knife could machine some brittle and hard materials [61]. In this work, the machined material is polymethyl methacrylate (PMMA), used for its moderate optical properties.

Cutter-path planning has the most significant effect on machining proper surface profiles. In traditional mechanical milling machining, the machining direction of the knife-edge and machining path is orthogonal when removing or cutting materials. However, for scraping materials using a microshaper, the machining path and machining direction of the knife-edge are the same. Therefore, a cutter-path planning algorithm used for microscraping needs to be developed. For cutter-path planning, the machined surfaces would be considered parametric or nonparametric surfaces. Due to the cutting behaviors of scraping being different from milling, a nonparametric surface is more suitable for the path planning used in scraping using the microshaper [47]. In nonparametric path-planning, iso-plane, iso-level, iso-scallop, and iso-parametric methods are developed [56]. Due to using a three-axis CNC machine, the iso-plane cutter-path planning method is chosen for this work. This method is also suitable for the specific cutting characteristics of the microshapers.

For the goal of a simple, easy, inexpensive, and quick machining method, the microshaper is mounted on an inexpensive three-axis CNC machine. As mentioned above, the algorithm of cutter-path planning is significant for achieving precise machining profiles. To derive the algorithm of cutter-path planning, the model is expressed in Figure 2. The feed direction is defined as *x*-coordinate. When the knife moves in the *x*-direction, the knife also changes the scraping height (*z*-direction) synchronously. Therefore, the curve in each path of the surface is machined. After machining in this path is finished, the knife is raised and moved to the next adjacent path transversely along the *y*-direction. When the knife is located at the beginning of the cutter path, the CNC machine will change the height of the knife in the *z*-direction to a specific position calculated using the cutter-path planning algorithm to execute the subsequentmachining. Furthermore, due to the mechanism of the CNC machine, the assigned movements are called step size in the transverse (or path interval) direction Δ*y* and step size in the cutter-path direction Δ*x*, respectively. The step sizes are determined geometrically by line segments that are used to match curves, tip the radius of the knife nose, and the surface curvatures and slopes, etc. Due to the iso-plane method being used for path planning in this work, the cutting surface will be expressed mathematically as:(1)z=z(x,y)

It is known that each path on the surface is a curve. However, the machining trajectory of the knife is a line segment. This curve is combined with several line segments. Therefore, the geometric relationship between the designed curve of machined surface profiles and the knife dimensions is very important in cutter-path planning. Due to the curve being machined using the knife with line segment trajectory, the cutter will move following a series of steps that are composed of line segments. Therefore, the step sizes in cutter-path planning must be expressed mathematically by the geometric relationship between the dimensions of the knife and the designed machine profile. The step size Δ (Δ*x* or Δ*y*) can be mathematically derived as shown in Figure 3. As shown in Figure 3, the relationship between line segment and radius of curvature of the machined curve profile is
(2)$$l=2\rho \sin\frac{\alpha }{2}$$

$\rho =\frac{{\left[1+{\left(\frac{dz}{dx}\right)}^{2}\right]}^{\frac{3}{2}}}{\frac{d^{2}z}{dx^{2}}}$ in *x*-*z* plane and $\rho =\frac{{\left[1+{\left(\frac{dz}{dy}\right)}^{2}\right]}^{\frac{3}{2}}}{\frac{d^{2}z}{dy^{2}}}$ in *y*-*z* plane. Where, *ρ* is the radius of the curvature of the designed curved surface in *x-z* or *y-z* plane, *α* is the central angle between two steps, *l* is the line segments which compose the curved profiles of the machining surface, and *S_i_* is the contact point of the knife and machined material in the *i*th step. The step size can be expressed as:(3)$$\Delta =l\cos\theta =l\cos{\left(\tan^{-1}\left(\frac{dz}{dx}\right)\right)}_{\mathrm{in}\;x-z\;\mathrm{plane}}=l\cos{\left(\tan^{-1}\left(\frac{dz}{dy}\right)\right)}_{\mathrm{in}\;y-z\;\mathrm{plane}}$$

Furthermore, due to the dimensions of the knife, there will be interference between the knife and the design surface, as shown in Figure 4. Therefore, it is necessary to compensate for knife height (*z*-direction) in cutter-path planning. The compensation of knife height can be executed by the cutter offset as shown in Figure 5. As shown in Figure 5, the designed cutting point is S due to calculating from step size Δ. However, interference could happen. Therefore, the cutting point in the microshaper must be point *C*. Moreover, the knife must be raised until the tip of the knife’s nose reaches point *L*. From Figure 5, the cutter offset needed to avoid interference is calculated:(4)$$\delta =\delta_{1}+\delta_{2}$$
where, $\delta_{1}=r(\frac{1}{\cos\theta }-1)$, $\delta_{2}=\frac{\rho }{\cos\theta }(\frac{1}{\cos\alpha }-1)$, $\theta =\tan^{-1}\left(\frac{dz}{dx}\right)$ in *x**-z* plane, $\theta =\tan^{-1}\left(\frac{dz}{dy}\right)$ in *y-z* plane, and from Equation (2) $\alpha =\tan^{-1}\left(\frac{r}{\rho }\tan\theta \right)$.

Interference is not only happening in the profile of the curved surface (*x-z* and *y-z* plane) but also in the boundary of the surface (*x-y* plane). As shown in Figure 6, if the designed boundary of the curved surface is a circle with radius *R*_0_ (black curve in the figure), the cutter path needs offsetting to compensate for the cutter interference. Therefore, the practical boundary must become the boundary with cutter offsetting (the red curve in this figure). As shown in Figure 6, the compensation of path direction (*x*-direction) *δ_x,i_* is:(5)$$\delta_{x,i}=\frac{\delta_{x}^{*}}{\frac{dz}{dx}}\;\mathrm{where}\;\delta_{x}^{*}=\sqrt{{\left(R_{0}+r\right)}^{2}-y^{2}\left(i\right)}-\sqrt{R_{0}^{2}-y^{2}\left(i\right)}\;\mathrm{and}\;y\left(i\right)=\sum \Delta y\left(i\right)$$

Moreover, the compensation in path interval direction (*y*-direction) happening in path 0 and is denoted as *δ_y,_*_0_:(6)$$\delta_{y,0}=\frac{\delta_{y}^{*}}{\frac{dz}{dy}}\;\mathrm{where}\;\delta_{y}^{*}=r\left(1-\cos\theta \right)$$

In traditional CNC machining with path planning, the complex surfaces could be machined. For different surfaces, conic section surfaces are significant due to their mathematic properties and wide applications. There are many applications in the macro world using conic surfaces. For example, elliptic surfaces can be used in aspheric lenses. Parabolic surfaces can be used to reflect the LED light. Furthermore, hyperbolic surfaces are used in the heat dissipation of cooling towers. Therefore, the revolution of conic section profiles is chosen as the curved surface of aspherical lenses to verify the feasibility of the algorithm. If the surface is a revolution surface conic profile, the implicit surface can be expressed as:(7)$$z=\frac{sq^{2}}{R+\sqrt{R^{2}-(1+k)q^{2}}}+a_{1}q^{2}+\cdot \cdot \cdot +a_{n}q^{2n}+\cdot \cdot \cdot$$
where, $q=\sqrt{x^{2}+y^{2}}$ is the distance measured from the central axis, *R* is the radius of curvature at the tip point of this revolution surface, *k* is the conic constant, and *a*_1_, …, and *a_n_* are the coefficients. If *s* = +1, this surface is concave. If *s* = −1, this surface is convex. The types of conic sections are determined by the conic constant *k*. Figure 7 shows different conic sections related to the values of *k* of Equation (7). From Figure 7, if *k* = −1, the surface is a paraboloid. If *k* < −1, the surface is a hyperboloid. If *k* > −1, the surface is ellipsoid. For micromachining of the conic section profiles, the high-order terms of Equation (7) are small enough to be neglected. Equation (7) can be expressed as:(8)$$z=\frac{sq^{2}}{R+\sqrt{R^{2}-(1+k)q^{2}}}$$

Due to the implicit function of the conic section being expressed as Equation (8), the step size can be derived as shown in Appendix A.

## 3. Results

For fabricating the micro-aspherical lenses, the microshaper is assembled on a three-axis vertical milling machine (YEONG CHIN CNC-50, Taichung, Taiwan). A 1 mm-thick PMMA substrate was used to manufacture the curved surfaces of the micro-aspherical lenses on this substrate. Figure 8 illustrates the cross-sections of the micro-aspherical lenses with different convex conic sections manufactured by the microshaper. Figure 8a illustrates the cross-section of the oblate ellipsoid microlens. Its conic constant *k* is 10 and the radius of the curvature at the tip point is *R* = 3 mm. Figure 8b displays the cross-section of the paraboloid microlens with *R* of 1 mm and *k* of −1. Figure 8c is the hyperboloid cross-section microlens with *R* = 1 mm and *k* = −20.

The profiles of the conic sections are measured on the microscope with a scale grid on the eyepiece. Figure 9 shows the image of the conic section observed on the microscope with a scale grid on the eyepiece. We add a hyperboloid to compare the profile of the machined convex surface. It shows that the profile of the microsurface is well-enveloped by the hyperboloid.

## 4. Discussion

Figure 10 shows the profile of the machined surface compared with the theoretical profile calculated from Equation (8). The profile of the machined surface is measured in the image saved from the microscope with a scale grid on the eyepiece, as shown in Figure 9. Figure 10a shows the machined oblate elliptic curve compared to the theoretical profile with *R* = 3 mm and *k* = 10. Figure 10b,c illustrate the comparison of the machined parabolic curves of *R* = 1 mm and *k* = −1 and hyperbolic curves of *R* = 1 mm and *k* = −20 with theoretical curves, respectively. From the results, it is demonstrated that the profiles of the three conic sections (oblate ellipsoid, paraboloid, and hyperboloid) machined by the microshaper agree well with the theoretical profiles. The microscraping method with cutter-path planning can machine precisely curved profiles. Therefore, it is proven that the aspherical lenses with specific profiles can be successfully fabricated and have good machined properties. It is noticed that the radius of the knife nose used in this work is larger than the reference [31]. However, the results show that the machining profiles precisely match the designed profiles. The significance of the cutter-path planning algorithm used in micro-aspherical lens manufacturing is verified.

Roughness is another important parameter that will affect the preciseness of machined profiles. Due to the characteristics of CNC machining, there should be scallop-inducing rough surfaces. Therefore, the machined surface needs further polishing. A surface roughness (TR-200D, Beijing TIME High Technology Ltd., Beijing, China) measurement instrument was used to measure the roughness of the machined surface. The roughness in the *y*-direction is larger than in the *x*-direction. The average roughness *R_a_* without polishing in *x*-direction and *y*-direction are 143 and 346 nm, respectively. It is noticed that the roughness magnitude in the *y*-direction is three times that of the value in the *x*-direction. The anisotropic roughness is due to the larger scallop happening in the *y*-direction during surface machining as shown in Figure 2. This magnitude of roughness could be a little large for the requirements of optical devices. Therefore, further polishing is needed to reduce the roughness. For PMMA, it can be polished using a simple method. Only by spreading toothpaste on a soft cloth and polishing the surfaces, will the roughness reduce to 44 nm and 52 nm in *x*- and *y*-direction, respectively. It is also noticed that these magnitudes of roughness are much smaller, and the roughness becomes isotropic. Figure 11 shows the profiles of the lens before and after polishing with toothpaste. It shows that before polishing, the profile has some scallops and that after polishing the profile becomes smoother.

It is known that aspherical lenses can compensate for aberration to obtain better images [44]. The image comparison between aspherical and spherical microlenses is discussed. Figure 12 shows the setup to observe image formation. The results are illustrated in Figure 13. Figure 13a is the sample image without a lens observed on the microscope. Figure 13b illustrates the image formed by the microspherical lens (*R* = 4 mm, *k* = 0) on the microscope. Figure 13c is the image formed by the micro-ellipsoid lens (*R* = 4 mm, *k* = −0.5) on the microscope. Figure 13d is the image formed by the micro-paraboloid lens (*R* = 4 mm, *k* = −1) on the microscope. It shows that the aspherical microlenses have better images than the spherical microlenses. Particularly on the edges of the lenses where the aberration would happen, the images formed by the aspherical microlenses have smaller distortions than the images formed by the spherical microlenses (as indicated by the red circle in Figure 13b).

## 5. Conclusions

A microshaper mounted on a CNC machine is used to fabricate microconvex aspherical lenses. For precisely scraping convex aspheric surfaces, cutter-path planning is exerted. The cutter-path planning algorithm with interference compensation is developed. Three convex aspherical microlenses with conic section (ellipsoid, paraboloid, and hyperboloid) surfaces are successfully fabricated. The machined profiles of the aspherical microlenses by the microshaper agree well and precisely with the designed theoretical conic section curves. The machined roughness can quietly diminish by using a simple polishing method. The roughness could reduce from 143 and 346 nm before polishing to 52 and 44 nm in path line direction and path interval direction, respectively. The image results show that the micro-aspherical lenses have better image formation than the microspherical lenses. The feasibility of precisely machining a designed profile with a specific function using the cutter-path planning algorithm developed in this work has been proven.

## Figures and Tables

**Figure 1 micromachines-13-00778-f001:**
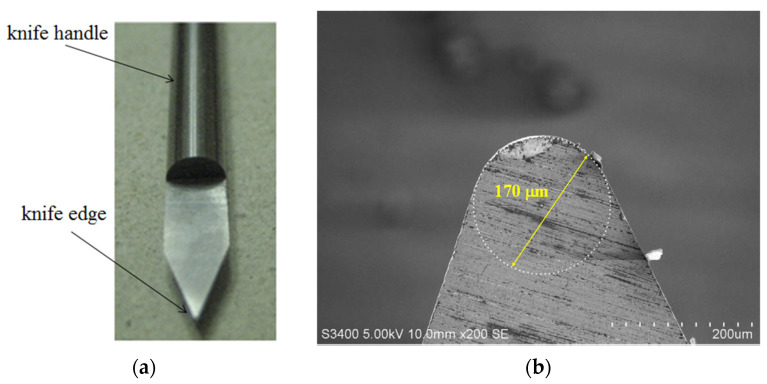
The microshaper mounted on the CNC machine has a small nose radius on the knife edge. (**a**) Photo of the microshaper. (**b**) SEM of the knife tip of microshaper. The tip of knife has diameter of 170 μm (radius of 85 μm).

**Figure 2 micromachines-13-00778-f002:**
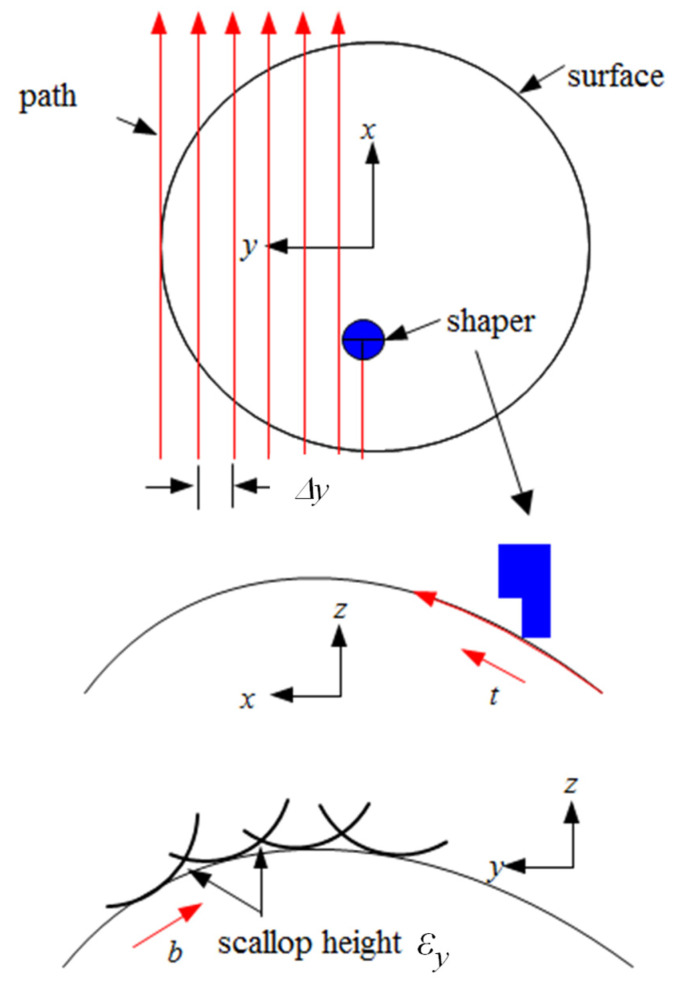
Expression of curved surface machined by the microshaper.

**Figure 3 micromachines-13-00778-f003:**
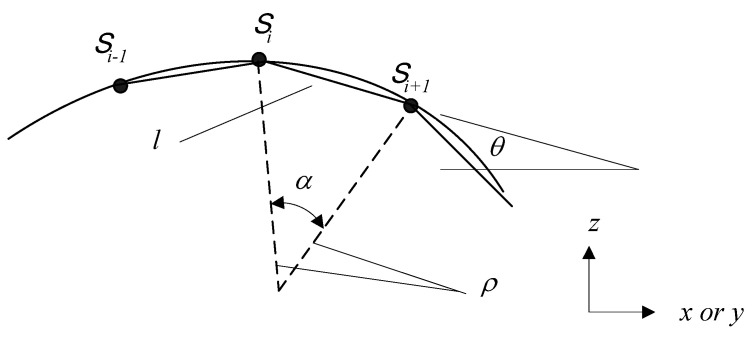
Geometry relationship between surface curve and microshaper path line segments.

**Figure 4 micromachines-13-00778-f004:**
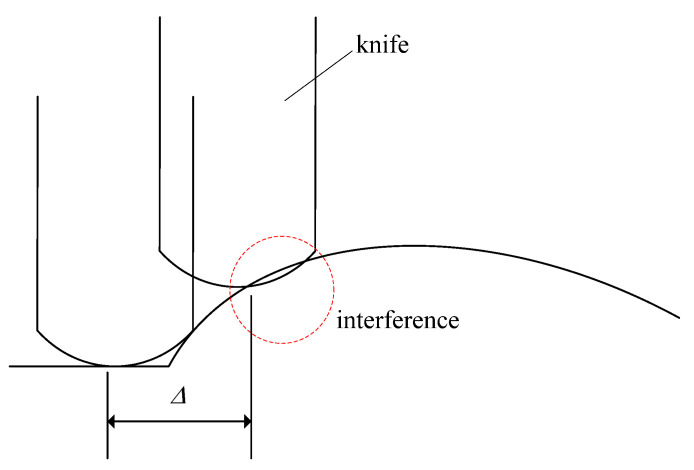
Interference of knife and designed surface.

**Figure 5 micromachines-13-00778-f005:**
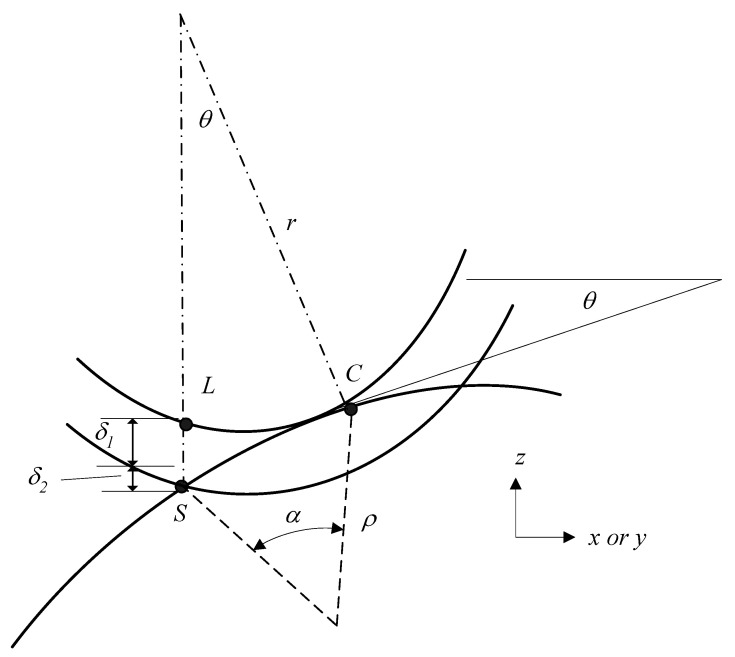
Geometry compensation of surface profile using cutter offsetting.

**Figure 6 micromachines-13-00778-f006:**
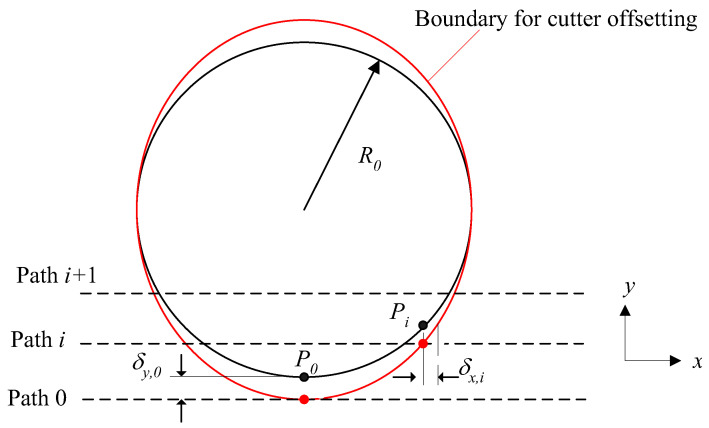
Boundary compensation using cutter offsetting.

**Figure 7 micromachines-13-00778-f007:**
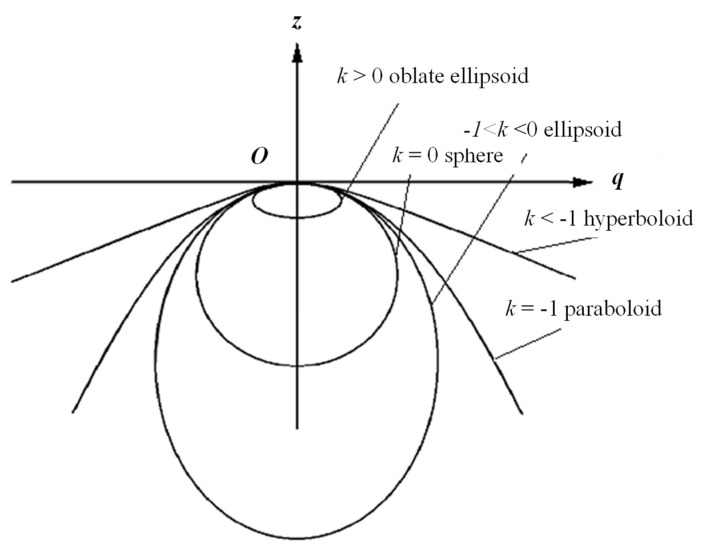
Expression of conic sections and their related conic constants.

**Figure 8 micromachines-13-00778-f008:**
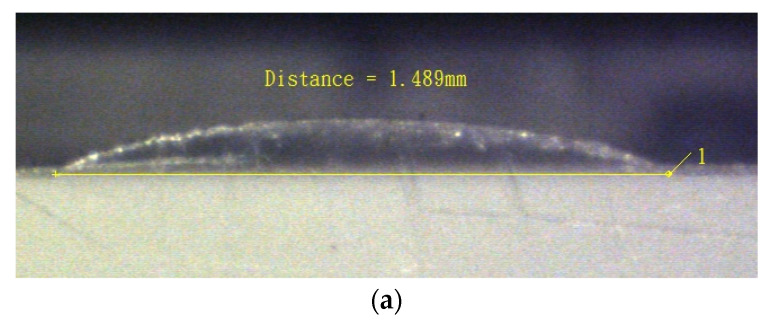
Cross section of conic section of micro-aspherical lenses fabricated by the microshaper scraping. (**a**) Oblate ellipsoid (*R* = 3 mm, *k* = 10), (**b**) paraboloid (*R* = 1 mm, *k* = −1), and (**c**) hyperboloid (*R* = 1 mm, *k* = −20).

**Figure 9 micromachines-13-00778-f009:**
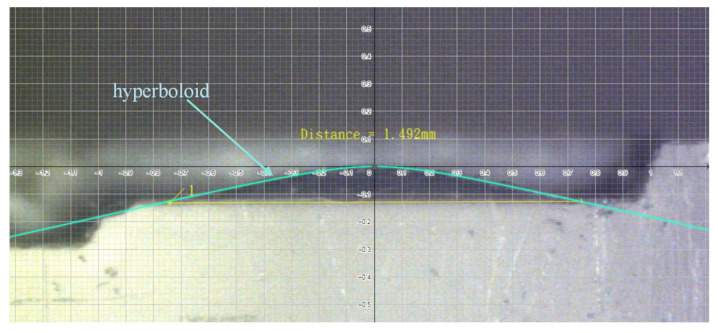
Conic section observed on microscope with scale grid on eyepiece.

**Figure 10 micromachines-13-00778-f010:**
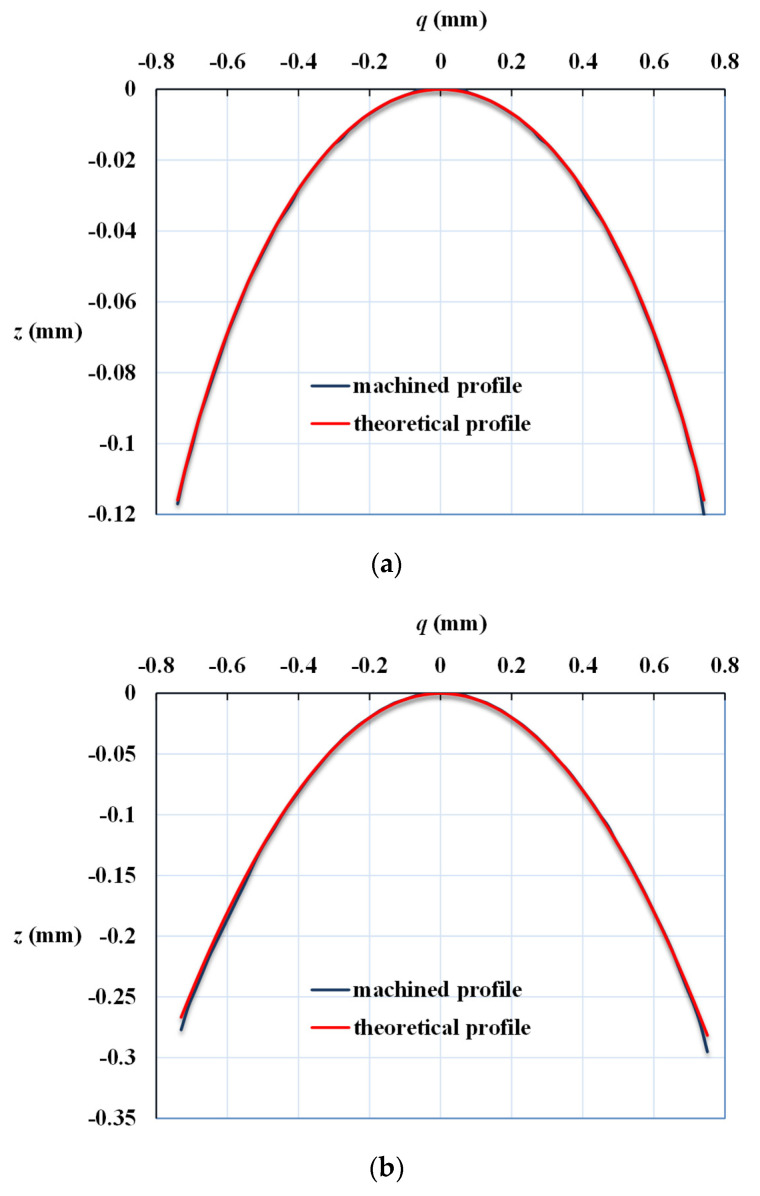
Comparison of machined curved surface profiles with theoretical conic section profiles calculated from Equation (8). (**a**) Oblate ellipsoid (*R* = 3 mm, *k* = 10), (**b**) paraboloid (*R* = 1 mm, *k* = −1), and (**c**) hyperboloid (*R* = 1 mm, *k* = −20).

**Figure 11 micromachines-13-00778-f011:**
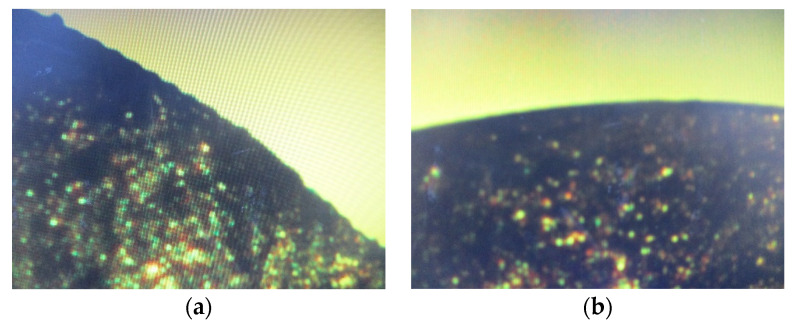
Images of cross-section of micro-asperical lenes (**a**) before polishing and (**b**) after polishing.

**Figure 12 micromachines-13-00778-f012:**
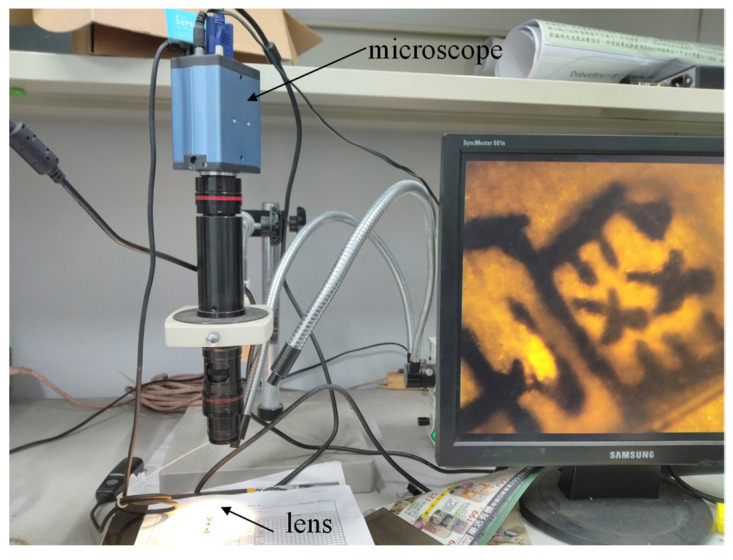
Setup of observing image formation by the microlens.

**Figure 13 micromachines-13-00778-f013:**
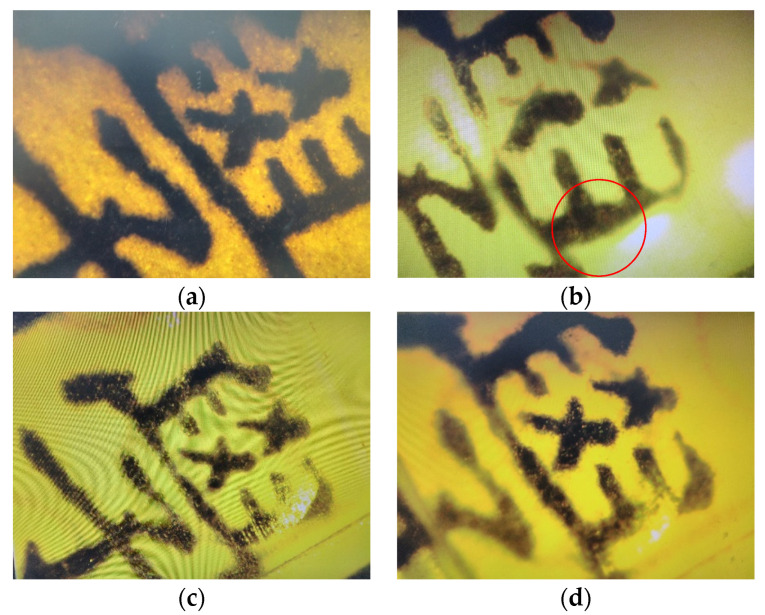
Image comparison. (**a**) Sample image observed on the microscope without a lens, (**b**) image formed by the microspherical lens (*R* = 4 mm, *k* = 0) on the microscope, (**c**) image formed by the micro-aspherical lens (*R* = 4 mm, *k* = −0.5) on the microscope, (**d**) image formed by the microspherical lens (*R* = 4 mm, *k* = −1) on the microscope.

## Data Availability

Not applicable.

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
