# Peer review of "Micro Aspheric Convex Lenses Fabricated by Precise Scraping"

_micromachines, 2022, doi:10.3390/mi13050778_

Round 1
Reviewer 1 Report
In this work, the author demonstrates an easy, fast, inexpensive method to fabricate micro convex aspherical lenses by utilizing a microshaper tool in conjunction with a CNC machine. The author mathematically determines the shapes of the aspheric surfaces, programs them into the CNC machine, and analyzes the final result.
Issues with the paper are listed below.
- The author presents many other fabrication techniques including reflow and laser etching. Given the abundance of similar work in the literature, they should place their results side-by-side with existing published results and give a statistical analysis of their technique in comparison.
- The author attempts to discuss material properties of the microshaper and the surfaces that it is cutting, using statements like, "the microshaper is tungsten steel which has enough hardness and stiffness". However, this statement is qualitative, and does not make a scientific comparison between the material properties of the microshaper and the material being cut into. To make this statement, the author should discuss numerical values of the relative hardness and stiffness compared to other materials they mention, like metal, polymers, and plastic. They should also discuss how using a different non-PMMA substrate material will react to the microshaper.
- The author should provide a brief discussion on the wear and deformation of the microshaper tool after N number of cuts. How does the end user know if the geometry of the microshaper changes when it is cutting into a harder material (ex. metal)?
- The author compares their proposed work to their previous work (Lin, 2021), in which the radius of the microshaper tool is actually smaller than that of their current tool. It seems the smaller radius microshaper tool from their previous work would yield better detail and better cuts, but the author makes no mention of this in their results.
- The author points out that the surface roughness without polish are 143 and 346 nm, which is not sufficient for optical device requirements. The author then claims that "spreading toothpaste on soft cloth" can be used to polish the surfaces down to 44 and 52 nm along the X- and Y-axes respectively. Since these resources are undoubtedly more easy to come across than the microshaper tool itself, the author should show that the technique does indeed work on these micro-structures. Sufficient evidence would be before and after profilometer measurements of the pre- and post-toothpaste treatment.
- Sufficient evidence to support the author's claims of being able to polish the micro-shapes is even more crucial, given that it is mentioned multiple times in their paper, and is literally the final sentence in their paper.
- The author substantiates their current work in aspherical surfaces by stating that "spherical lenses have aberration problems and would cause blurry images". In fact, the author fabricated such spherical microlenses in their previous work (2021). In their previous work, the authors showed images of a laser passing through their fabricated design (Fig. 10). Given the author's claim that their current aspheric surfaces are superior to the aberration-causing spherical surfaces, it should be straight forward for the author to demonstrate that their new surfaces perform better than their previous designs. This is neither mentioned nor included in the current paper.
- Lastly, the grammar and spelling of the current paper need to be improved. Along with the grammatical errors and spelling mistakes, this paper also contains incomplete sentences, such as "Due to iso-plane method is used for path planning in this work."
In conclusion, this reviewer believes the work contained in this paper lacks the novelty and significant contributions to the scientific community needed to justify publication of this work. In their previous paper (2021), the author presents the fabrication of a spherical microlens array via the same method used in this paper. Generally speaking, the previous technique follows the same flow as this paper: (a) the author determines the shape they want to create, (b) feeds it into a CNC machine equipped with a microshaper, (c) the author analyzes the results.
In this sense, the current paper presents no new techniques to the field. The equations presented in their work for the mathematical description of aspherical surfaces (ex. ellipsoids, paraboloids, and hyperboloids) is well known to the field. Additionally, the authors actually present less data in this paper to substantiate their claims than they did in their previous 2021 publication, "Microlens Array Fabrication by Using a Microshaper"
Author Response
In this work, the author demonstrates an easy, fast, inexpensive method to fabricate micro convex aspherical lenses by utilizing a microshaper tool in conjunction with a CNC machine. The author mathematically determines the shapes of the aspheric surfaces, programs them into the CNC machine, and analyzes the final result.
Issues with the paper are listed below.
- The author presents many other fabrication techniques including reflow and laser etching. Given the abundance of similar work in the literature, they should place their results side-by-side with existing published results and give a statistical analysis of their technique in comparison.
My reply:
Thanks a lot for reviewer’s comments. I have added some expressions to the literal review as “For the aforestated silicon-based micromachining methods, they focus on the applications of micro lenses. Although the lenses with quasi-spherical shapes can be fabricated by physical characteristics, these methods are hard to control the precise profile of the lenses.” and “The aforestated works presented aspherical lenses can be fabricated by silicon-based micromachining. However, their aspherical profiles are formed by fabrication parameters and applying voltages. They have no designed functions of the lens profiles.”
- The author attempts to discuss material properties of the microshaper and the surfaces that it is cutting, using statements like, "the microshaper is tungsten steel which has enough hardness and stiffness". However, this statement is qualitative, and does not make a scientific comparison between the material properties of the microshaper and the material being cut into. To make this statement, the author should discuss numerical values of the relative hardness and stiffness compared to other materials they mention, like metal, polymers, and plastic. They should also discuss how using a different non-PMMA substrate material will react to the microshaper.
My reply:
Thanks a lot for reviewer’s comments. I increased references 60 and 61 to explain the hardness, stiffness, and wear resistance of tungsten steel (tungsten carbide) tool.
- The author should provide a brief discussion on the wear and deformation of the microshaper tool after N number of cuts. How does the end user know if the geometry of the microshaper changes when it is cutting into a harder material (ex. metal)?
My reply:
Thanks a lot for reviewer’s comments. I increased references 60 and 61 to explain the hardness, stiffness, and wear resistance of tungsten steel (tungsten carbide) tool.
- The author compares their proposed work to their previous work (Lin, 2021), in which the radius of the microshaper tool is actually smaller than that of their current tool. It seems the smaller radius microshaper tool from their previous work would yield better detail and better cuts, but the author makes no mention of this in their results.
My reply:
Thanks a lot for reviewer’s comments. The radius of knife nose used in this work is larger than previous work. And the smaller radius of knife tip would have better cuts for common sense. I add the expression “It is noticed that the radius of knife nose used in this work is larger than Reference [31]. But the results show machining profiles precisely matching the designed profiles. The significance of cutter path planning algorithm used in micro aspherical lenses manufacturing is verified.”
The author points out that the surface roughness without polish are 143 and 346 nm, which is not sufficient for optical device requirements. The author then claims that "spreading toothpaste on soft cloth" can be used to polish the surfaces down to 44 and 52 nm along the X- and Y-axes respectively. Since these resources are undoubtedly more easy to come across than the microshaper tool itself, the author should show that the technique does indeed work on these micro-structures. Sufficient evidence would be before and after profilometer measurements of the pre- and post-toothpaste treatment.
My reply:
- Thanks a lot for reviewer’s comments. I added Figure 11 to explain. And the roughness is quantitatively measured by surface roughness (TR-200D, Beijing TIME High Technology Ltd, Beijing, China) measurement instrument which is mentioned in this work.
- The author substantiates their current work in aspherical surfaces by stating that "spherical lenses have aberration problems and would cause blurry images". In fact, the author fabricated such spherical microlenses in their previous work (2021). In their previous work, the authors showed images of a laser passing through their fabricated design (Fig. 10). Given the author's claim that their current aspheric surfaces are superior to the aberration-causing spherical surfaces, it should be straight forward for the author to demonstrate that their new surfaces perform better than their previous designs. This is neither mentioned nor included in the current paper.
My reply:
Thanks a lot for reviewer’s comments. The comparison is shown on Figure 13. And I wrote the expression “It is known that aspherical lenses can compensate for aberration to obtain better im-ages [44]. The image comparison between aspherical and spherical microlenses is dis-cussed. Figure 12 shows the setup to observe image formation. The results are illustrated in Figure 13. Figure 13 (a) is the sample image without a lens observed on the microscope. Figure 13 (b) illustrates the image formed by the micro spherical lens (R = 4 mm, k = 0) on the micro-scope. Figure 13 (c) is the image formed by the micro ellipsoid lens (R = 4 mm, k = -0.5) observed on microscope. And Figure 13 (d) is the image formed by micro paraboloid lens (R = 4 mm, k = -1) shown on microscope. It shows that the aspherical microlenses have better images than spherical micro lenses. Especially on edge of lenses where the aberration would happen, the images formed by aspherical microlenses have smaller distortion than the image formed by spherical micro lens (as indicated by red circle in Figure 13 (b)).”
- Lastly, the grammar and spelling of the current paper need to be improved. Along with the grammatical errors and spelling mistakes, this paper also contains incomplete sentences, such as "Due to iso-plane method is used for path planning in this work."
My reply:
Thanks a lot for reviewer’s comments. I am sorry for my poor English. By asking for other’s help, I have checked the grammar and spelling and corrected the mistakes.
In conclusion, this reviewer believes the work contained in this paper lacks the novelty and significant contributions to the scientific community needed to justify publication of this work. In their previous paper (2021), the author presents the fabrication of a spherical microlens array via the same method used in this paper. Generally speaking, the previous technique follows the same flow as this paper: (a) the author determines the shape they want to create, (b) feeds it into a CNC machine equipped with a microshaper, (c) the author analyzes the results.
In this sense, the current paper presents no new techniques to the field. The equations presented in their work for the mathematical description of aspherical surfaces (ex. ellipsoids, paraboloids, and hyperboloids) is well known to the field. Additionally, the authors actually present less data in this paper to substantiate their claims than they did in their previous 2021 publication, "Microlens Array Fabrication by Using a Microshaper"
My reply:
Thanks a lot for reviewer’s comments. My previous work developed an easy, simple, quick and inexpensive method to fabricate micro spherical lenses. This work follows previous work machining method to fabricate micro aspherical lenses. However, it is not only extending of previous work. This work developed a cutter path planning algorithm and compensation of kike interference. It would help to machine any profiles with specific functions. The three conic section profiles are used to prove the feasibility of this method.

Reviewer 2 Report
This manuscript is well organized and could be accepted after minor revision.
The following issues should be concerned。
1.The innovation of this research cannot be extracted from the introduction section.
2. The theoretical findings cann't be found or concluded in the conclusion section, although some experimental results were given.
Author Response
This manuscript is well organized and could be accepted after minor revision.
The following issues should be concerned。
- The innovation of this research cannot be extracted from the introduction section.
My reply:
Thanks a lot for reviewer’s comments. I wrote two sentences to express the innovation:
“In this work, the cutter path planning algorithm is developed. The compensation algo-rithm to avoid knife interference is also derived.” and “Therefore, precisely machining a profile with specific function using the algorithm of cut-ter path planning developed in this work is feasible.”
- The theoretical findings cann't be found or concluded in the conclusion section, although some experimental results were given.
My reply:
Thanks a lot for reviewer’s comments. I wrote the sentences to express the theoretical findings of the algorithm: “The cutter path planning algorithm with interference compensation is developed.” and “The feasibility of precisely machining a designed profile with specific function by the cutter path planning algorithm developed in this work is proved.”

Round 2
Reviewer 1 Report
Corrections made by the author were sufficient to fill in gaps of the previous submission.